# Inferring ECG Waveforms from PPG Signals with a Modified U-Net Neural Network

**DOI:** 10.3390/s24186046

**Published:** 2024-09-19

**Authors:** Rafael Albuquerque Pinto, Hygo Sousa De Oliveira, Eduardo Souto, Rafael Giusti, Rodrigo Veras

**Affiliations:** 1Instituto de Computação, Universidade Federal do Amazonas (UFAM), Av. Rodrigo Otávio, n° 6200, Manaus 69077-000, AM, Brazil; hygo.sousa@icomp.ufam.edu.br (H.S.D.O.); esouto@icomp.ufam.edu.br (E.S.);; 2Departamento de Computação, Universidade Federal do Piauí (UFPI), R. Dirce Oliveira, n° 1805, Teresina 64049-550, PI, Brazil; rveras@ufpi.edu.br

**Keywords:** photoplethysmogram, electrocardiogram, U-net neural network, wearable devices, continuous monitoring

## Abstract

There are two widely used methods to measure the cardiac cycle and obtain heart rate measurements: the electrocardiogram (ECG) and the photoplethysmogram (PPG). The sensors used in these methods have gained great popularity in wearable devices, which have extended cardiac monitoring beyond the hospital environment. However, the continuous monitoring of ECG signals via mobile devices is challenging, as it requires users to keep their fingers pressed on the device during data collection, making it unfeasible in the long term. On the other hand, the PPG does not contain this limitation. However, the medical knowledge to diagnose these anomalies from this sign is limited by the need for familiarity, since the ECG is studied and used in the literature as the gold standard. To minimize this problem, this work proposes a method, PPG2ECG, that uses the correlation between the domains of PPG and ECG signals to infer from the PPG signal the waveform of the ECG signal. PPG2ECG consists of mapping between domains by applying a set of convolution filters, learning to transform a PPG input signal into an ECG output signal using a U-net inception neural network architecture. We assessed our proposed method using two evaluation strategies based on personalized and generalized models and achieved mean error values of 0.015 and 0.026, respectively. Our method overcomes the limitations of previous approaches by providing an accurate and feasible method for continuous monitoring of ECG signals through PPG signals. The short distances between the infer-red ECG and the original ECG demonstrate the feasibility and potential of our method to assist in the early identification of heart diseases.

## 1. Introduction

According to studies conducted by the World Health Organization (WHO), cardiovascular diseases have been one of the leading causes of death worldwide over the past 20 years [1]. More than half a billion people around the world continue to be affected by cardiovascular diseases, which accounted for 20.5 million deaths in 2021, close to a third of all deaths globally [2]. This increase can be attributed mainly to the sedentary lifestyle, work-related stress, obesity, and smoking habits of the population.

In general, the monitoring of cardiac activities is done in clinical environments, making diagnosing many cardiovascular diseases a challenging task, since different types of cardiac anomalies, such as intermittent arrhythmias, often occur outside the clinical environment, for instance, during physical activity or sleep. For this reason, continuous monitoring of cardiac vital signs is seen as essential to provide an early diagnosis of these diseases [3].

The 12-lead electrocardiogram (ECG) is commonly used in clinical settings and is considered the gold standard for measuring and monitoring the heart’s electrical activities. However, obtaining continuous ECG signals can be inconvenient due to the need for electrodes, wires, gel, adhesives, and metal pins placed on various parts of the body, which can restrict patient movement and cause skin irritation and discomfort during prolonged and continuous use [4,5]. Therefore, significant research efforts are underway to create wearable ECG devices suitable for daily use. However, progress in this area has been limited, and currently, only a few wrist devices (smartwatches and smart bands) offer ECG monitoring. Even these devices have limitations, as they require users to stand still and touch the watch with both hands to record a limited segment of ECG (typically only 30 s) [6].

Photoplethysmography (PPG) is an optical method that detects changes in blood volume in the microvascular layer of skin tissue with each heartbeat [7]. PPG is a simple and non-invasive alternative to ECG. Several studies have shown that several features extracted from the PPG signal (e.g., pulse rate variability) have a high correlation with corresponding metrics extracted from the ECG (e.g., heart-rate variability) [8].

Although there is a similarity in the data captured by each method (ECG and PPG), finding a function that maps PPG to ECG signals introduces new challenges regarding the system’s accuracy. It happens because the PPG signal alone is quite sensitive to noise and has a temporal misalignment with the ECG signal, mainly due to pulse transit time, which is the time it takes for the heartbeat wave to reach the extremities of the body [9].

ECG waveforms derived from PPG signals exhibit several distinct characteristics compared to ECG data obtained directly. PPG signals are highly sensitive to noise, especially motion artifacts, ambient light interference, and pressure variations, which can introduce irregularities in the inferred ECG waveforms, making accurate reconstruction challenging. Due to pulse transit time, there is a temporal misalignment between PPG and ECG signals. This misalignment can cause discrepancies in the temporal characteristics of the reconstructed ECG waveforms.

The morphological features of ECG waveforms derived from the PPG can vary due to the intrinsic differences in measurement mechanisms. While the ECG directly measures the heart’s electrical activity, the PPG measures it indirectly through blood volume changes, leading to potential distortions in the reconstructed waveforms. Despite these differences, studies have shown that various metrics extracted from a PPG, such as heart-rate variability (HRV), have a high correlation with corresponding ECG metrics. With appropriate signal processing and machine learning techniques, reasonably accurate ECG waveforms can be inferred from a PPG.

Despite these challenges, mapping a PPG to an ECG is highly advantageous because it enables continuous and non-invasive cardiac monitoring, which can be particularly beneficial in remote or ambulatory settings. This capability can lead to the earlier detection of cardiac anomalies, more timely interventions, and, ultimately, better patient outcomes. Furthermore, the convenience and comfort of PPG-based monitoring can improve patient compliance, providing more comprehensive and accurate long-term cardiac health data.

This work proposes a method for reconstructing ECG signals from PPG signals using a convolutional neural network architecture (CNN) to overcome the limitations presented by existing methods. The proposed network architecture is based on the U-Net network [10], which is commonly used for 2D medical image segmentation applications [11]. For this work, the network architecture was adapted by changing the number of layers and replacing the 2D convolution operations with their 1D counterparts.

Our previous work [12] introduced a deep learning network designed in an encoder–decoder topology using fully convolutional blocks. The model was optimized to map between the signals, enabling continuous monitoring of cardiac activity via PPG.

In this current work, inception residual blocks were introduced to enhance the parallel processing power of CNNs. These blocks facilitate parallel convolutions of varying filter sizes, enabling the neural network to capture information at multiple scales. Skip connections between the encoder and decoder paths were implemented to preserve high-granularity spatial information, improving overall network performance.

This paper also explores the following research questions:Group Model (GM): Can a model using training data from all individuals (generic model) predict ECG waveforms from unseen PPG measurements for an individual?Subject-Specific Model (SM): Can a model trained with the majority of an individual’s ECG and PPG signal data predict their ECG waveforms from unseen PPG measurements for the same individual?

Our models were evaluated using four public ECG–PPG datasets containing 125 individuals with a wide range of ages, weights, and health conditions. The results demonstrate that the proposed models can accurately reconstruct ECG signals from PPG signals when trained in a generic or personalized manner.

The main contributions of our study are presented below:We propose a novel method called PPG2ECG for reconstructing ECG signals from PPG signals, specifically tailored to address the inherent challenges of noise and morphological variations in the input signals. To achieve this, we applied a pre-processing filter to enhance signal clarity and mitigate noise. More importantly, we introduced significant modifications to the IncResU-Net architecture. Our key contribution is the adaptation of the IncResU-Net architecture by incorporating 1D convolutions, which are better suited for processing temporal physiological signals such as a PPG and an ECG. Additionally, we integrated dilated residual blocks to effectively capture multiple temporal scales, enabling the model to better detect intricate variations in the signals over time. These architectural improvements result in more precise and realistic ECG signal reconstructions, addressing a gap in existing models that often struggle with the complexity and variability of physiological signals.We conduct a comprehensive evaluation of the proposed PPG2ECG method using four diverse datasets, comparing its performance in terms of similarity and reconstruction error with two state-of-the-art methods. Our evaluation covers datasets that represent a wide range of conditions, ensuring a robust analysis of the model’s effectiveness. By benchmarking against leading methods, we demonstrate that our approach not only matches but, in several key aspects, surpasses current techniques for ECG signal reconstruction from PPG signals.We perform a subject-independent study, demonstrating the generalization capability of the generated models to unseen individuals’ data acquired under different conditions. This evaluation underscores the versatility and robustness of our method, as it consistently produces accurate reconstructions even when applied to data from individuals not included in the training set. Such a generalization is critical for real-world applications, where models must perform reliably across diverse populations and under varying physiological and environmental conditions.

The rest of the article is organized as follows: Section 2 briefly mentions previous studies on ECG signal reconstruction. Next, our proposed method is discussed in Section 3. Section 4 discusses the details of our experiments, including training procedures, evaluation metrics used, and results and analysis. Finally, we conclude this article in Section 5.

## 2. Related Works

In the literature, various studies have utilized data extracted from PPG and/or ECG signals to identify and classify different heart pathologies. For instance, Kutlu and Kuntalp Kutlu and Kuntalp [13] proposed using k-Nearest Neighbors(kNN) to recognize and classify cardiac arrhythmias. Kumar et al. [14] employed support vector machines in characterizing arterial diseases. Hannun et al. [15] and Elhaj et al. [16] used deep learning models for detecting cardiac arrhythmias. However, despite the intrinsic correlation between PPG and ECG signals, more solutions still need to be developed for reconstructing ECG signals from PPG signals.

Banerjee et al. [17] suggested a method for inferring parameters of the ECG signal’s RR (the interval between two consecutive R-wave peaks), PR (the time from the onset of the P wave to the start of the QRS complex), QRS (the duration of the QRS complex representing ventricular depolarization), and QT (the interval from the beginning of the QRS complex to the end of the T wave, representing ventricular depolarization and repolarization) wave intervals from features extracted by observing the time and frequency domains of the PPG signal. They used a multilayer perceptron and support vector machine to make these inferences. Although the method’s performance achieved 90% accuracy in estimating the ECG parameters, the limited ability to infer only some ECG parameters restricted the broader use of this technique.

Zhu et al. [4,18] analyzed the relationship between the ECG and PPG signals using a transformation technique that maps the discrete cosine transform (DCT) coefficients of each PPG cycle to the corresponding ECG cycle. The resulting DCT coefficients from the PPG cycle were inversely transformed to obtain the reconstructed ECG waveform. Experimental results showed that the proposed method can achieve high accuracy of 0.92 in mean correlation.

Tian et al. [19] presented a dictionary learning-based method to find a sparse representation of the input PPG data as a linear combination with the output ECG data. To perform the mapping, the PPG and ECG signals were pre-processed into normalized signal cycles. The proposed model was evaluated with over 34,000 ECG/PPG cycle pairs containing a variety of ECG morphologies and cardiovascular diseases. The experimental results validated the accuracy and generality of the proposed algorithm, achieving a mean reconstruction correlation of 0.96.

More recently, some studies have proposed using deep neural networks for reconstructing ECG signals from PPG signals. Sarkar and Etemad [6] proposed a method based on generative adversarial networks (GANs) that employs attention mechanisms to focus on specific regions of the ECG signal, such as the QRS complex. Chiu et al. [20] presented a method composed of a sequence transformer network to address the problem of temporal shift in different signals, an attention network to learn the most important points of the input PPG signal for ECG reconstruction, and a CNN encoder–decoder to map the output of the first two components to the ECG signal.

Previous research on ECG signal reconstruction from PPG data has predominantly relied on deep learning architectures, such as those proposed by Sarkar and Etemad [6] and Chiu et al. [20]. While these methods have proven effective in capturing complex temporal characteristics, they face limitations, particularly regarding overfitting when applied to smaller datasets or noisy signals. Additionally, many of these approaches struggle to generalize across diverse patient populations, as ECG signals exhibit significant morphological variation influenced by factors such as age, gender, health conditions, and anatomical differences.

In contrast, this study introduces the PPG2ECG method, which is designed to enhance generalization and efficiency by employing a one-dimensional convolutional neural network (CNN) based on an encoder–decoder architecture. One-dimensional convolutions are better suited for processing temporal physiological signals like a PPG and an ECG, allowing the model to efficiently capture the dynamic nature of these signals. Furthermore, by incorporating dilated residual blocks, the network can capture multiple temporal scales, enabling it to detect subtle variations in the signal over time. These architectural choices allow for more accurate and realistic ECG reconstructions, addressing the limitations of previous models, which often struggled with the complexity and variability of physiological data.

## 3. Proposed Method

Creating a model to reconstruct ECG signals from PPG signals involves three main stages, which are illustrated in Figure 1. Former data were collected from PPG and ECG sensors. Then, the signals were pre-processed, including filtering, normalization, and segmentation over time. Finally, a regression model was trained using an adapted version of the IncResU-Net architecture (combination of two architectures Inception-ResNet and U-Net) [11]. The following sections provide more information about each stage of the PPG2ECG method.

### 3.1. Dataset Description

In this study, we combined four ECG-PPG datasets: BIDMC [21], CAPNO [22], DALIA [23], and WESAD [24]. The combined dataset contains 125 participants with a balanced ratio of males and females. In all datasets, PPG and ECG signals were recorded simultaneously. The waveform shapes of the PPG and ECG signals vary depending on the databases, as some were collected in controlled environments using clinical devices, while others were collected during daily activities using wearable devices. Therefore, the dataset with the combined databases has a wide variety of noisy and abnormal signals, requiring the application of signal processing techniques. Since the datasets were collected by different devices, each dataset presented varying sampling frequencies, making it necessary to perform a resampling process on the ECG and PPG signals to unify the sampling rate. After the resampling process using interpolation, the aggregated dataset was standardized to a sampling frequency of 128 Hz.

#### 3.1.1. BIDMC

The BIDMC dataset [21] is a subset of data extracted from the MIMIC-II dataset [25], consisting of ECG, PPG, and respiratory impedance pneumography (IP) signals from patients admitted to the medical and surgical intensive care units at Beth Israel Deaconess Medical Center (BIDMC) in Boston, MA, USA. The dataset comprises 53 randomly selected 8 min records of ECG, PPG, and IP signals acquired simultaneously from patients aged 19 to 90 years, all with a sampling frequency of 125 Hz.

#### 3.1.2. CAPNO

The CapnoBase TBME RR benchmark dataset [22] was designed to develop and test algorithms for estimating respiratory rate. It includes ECG, PPG, and capnography (CO_2_) signals from patients during elective surgeries and routine anesthesia, with records of spontaneous or controlled breathing. CapnoBase consists of 42 randomly selected 8 min recordings from a larger dataset, including 29 pediatric patients with a median age of 8.7 years and 13 adults with a median age of 52.4 years. The signals were collected with a sampling frequency of 300 Hz.

#### 3.1.3. DALIA

The PPG-DaLiA dataset [23] is a multimodal dataset of physiological and movement data created for motion compensation and heart rate estimation during daily activities such as walking, sitting, and driving. The raw data were collected using two commercial devices: the Empatica E4 worn on the wrist (PPG, 3-axis acceleration, electrodermal activity, and body temperature) and the RespiBAN Professional worn on the chest (ECG, respiration, and accelerometer). The dataset includes 2 h recordings of 15 participants, eight women and seven men, with an average age of 30.6. The signals collected from the wrist have a sampling rate of 64 Hz, while those collected from the chest device have a sampling rate of 700 Hz.

#### 3.1.4. WESAD

The Wearable Stress and Affect Detection (WESAD) dataset, introduced in [24], is a multimodal dataset that includes physiological and movement data. The dataset was created to detect stress and affect while performing activities such as solving arithmetic tasks and watching video clips. The raw data were captured using two commercial devices: the Empatica E4 wrist-worn device (which measures PPG, three-axis acceleration, electrodermal activity, and body temperature) and the RespiBAN Professional chest-worn device (which measures ECG, respiration, and acceleration). The dataset contains recordings from 15 participants, with over 1 h of data for each participant. The participants included three women and 12 men, with an average age of 27.5 years. The signals collected from the wrist have a sampling rate of 64 Hz, while the signals collected from the chest have a sampling rate of 700 Hz.

### 3.2. Data Pre-Processing

There are inherent issues with device technology and usage that can affect the quality of the data collected, such as motion artifacts, instrumental errors, electrical interference from other sources, and human errors. Gathering a large amount of data is also necessary to encompass greater diversity in the morphology of waves, since ECG and PPG signals come from patients with various clinical situations. Although many machine learning algorithms are designed to handle data in these situations, they may produce more accurate results if some of the issues present in the data are resolved. The following sections detail the signal processing techniques adopted to improve the quality of ECG and PPG signals.

#### 3.2.1. Signal Filtering

As mentioned earlier, ECG and PPG sensor readings can be contaminated by various types of noise, which may originate from either the body’s movements or external sources, such as electronic equipment or the power grid. In this way, filtering methods must be applied to eliminate these noises. Due to the complexity and diversity of application scenarios, the PPG signal suffers stronger interference than the ECG signal, which usually has less noise since it is typically captured in clinical settings. The literature suggests different filters for PPG and ECG signals due to the nature of the noise. According to [26], the Butterworth filter is considered to be the best filter to be applied to PPG signals. Its application is commonly used to remove baseline oscillation and high frequencies that usually affect the PPG signal waveform. Thus, applying the Butterworth filter aims to minimize changes in the PPG signal waveform that are not of cardiac origin. The contaminated PPG signal and noise can be represented as follows:(1)$$y^{\prime }\left(t\right)=y\left(t\right)+n\left(t\right),$$
where $y^{\prime }\left(t\right)$ is the contaminated signal, y(t) is the original PPG signal, and n(t) is the internal or external noise.

To eliminate the noise, the Butterworth filter can be applied to the $y^{\prime }\left(t\right)$ signal as follows:(2)$$y_{filt}\left(t\right)=F\left(y^{\prime }\left(t\right)\right),$$
where *F* is the Butterworth filter, and $y_{filt}\left(t\right)$ represents the PPG signal free of noise.

#### 3.2.2. Normalization and Segmentation of ECG and PPG Signals

After the filtering process, z-score normalization was applied to the entire time series of ECG and PPG signals in each sample to adjust them to a normal distribution. Z-score normalization considers the mean μ and standard deviation σ of a signal, which is obtained by the following formula:(3)$$x^{\prime }=\frac{y_{filt}\left(t\right)-\mu }{\sigma },$$
where the filtered original signal is denoted by $y_{filt}\left(t\right)$, and the normalized signal is designated by $x^{\prime }$. The normalization parameters were computed separately for each signal. Furthermore, the mean of each normalization parameter in the training stage was used to normalize all PPG signals in the prediction stage. Subsequently, the data with z-score normalization were fed into the segmentation stage described below. After the segmentation process, the data were normalized again to ensure that all input data for the neural model fell within a specific range. This was accomplished by applying Min-Max normalization to the segmented data (Equation (Equation 4)).
(4)$$x^{\prime }=\frac{\left(x-min(x)\right)}{\left(max(x)-min(x\right)},$$The segmentation process is responsible for splitting the time series into smaller sizes, which are defined as segments or time windows, so each segment has a limited number of samples according to size. In this study, each segment was defined with a length of 4 s. The size of the PPG and ECG signal segments was defined based on the signal sampling frequency. In this work, the segmentation process generated the same number of segments for both ECG and PPG signals for synchronization reasons of different signals. Therefore, the segmentation result will be a set of data, $X=\{\left(x^{\left(1\right)},y^{\left(1\right)}\right),\left(x^{\left(2\right)},y^{\left(2\right)}\right),\ldots ,\left(x^{\left(m\right)},y^{\left(m\right)}\right)\}$, which consists of the input PPG signal $x^{\left(i\right)}$ and the reference ECG signal $y^{\left(i\right)}$, where the input signal $x^{\left(i\right)}$ must be of the same size as the reference signal $y^{(i})$.

Each data segment $\left(x^{\left(i\right)},y^{\left(i\right)}\right)$ has an initial time $t_{i}$ and final time $t_{f}$, with a 10% overlap to avoid losing any peaks of the ECG and PPG signals. Figure 2 illustrates an example of the data’s formatting and graphical representation after the time series segmentation process obtained from the ECG and PPG sensors.

### 3.3. Network Architecture

The proposed network was inspired by the IncResU-Net network, which was used for 2D medical image segmentation applications [11]. The IncResU-Net network consists of a neural network architecture that combines the U-net [10] and Inception-ResNet [27] architectures. For this work, the architecture of the IncResU-Net network was adapted, with changes made to the number of layers and the two-dimensional convolution, pooling, and upsampling operations, which were replaced by their one-dimensional counterparts.

The proposed architecture consists of two stages: the encoder and the decoder, as illustrated in Figure 3. In the encoder stage, downsampling is performed using two input signals: the PPG and a channel of the ECG (derivative II). The encoder stage comprises four levels, where convolution operation is performed using filters of size 1 × 4, starting with 32 filters. Downsampling reduces the input size while increasing the number of filters at each level by a factor of two until the number of filters reaches 256. The choice of 1 × 4 filters in the early layers allows for capturing finer details in the temporal series of signals, focusing on the short-term temporal characteristics of the ECG signal. As the number of filters increases in the deeper layers, the model is able to learn more complex and abstract features, which are essential for accurately reconstructing the ECG from the PPG. This strategy of progressively increasing the number of filters improves the model’s ability to generalize and capture important patterns throughout the signal. This is followed by batch normalization, a ReLU (Rectified Linear Unit) activation function, and a dilated inception residual block. The batch normalization layer helps reduce covariance between the activations of hidden layers by transforming the input values to a zero-mean and unit variance distribution. The ReLU activation function, defined as f(x)=max(0,x), helps produce sparse activations by returning zero for negative input values and the input value for positive ones, thereby reducing the computational complexity of the model and improving its generalization capacity. The residual block enhances the network’s representation power. It makes it possible to capture complex features and patterns in the PPG signal, increasing the observation area without significantly increasing the network parameters, thereby helping to reduce the problem of vanishing gradients and improving convergence during training.

The upsampling operation is performed in the decoder stage using a deconvolution operation at each level, which is similar to the encoder stage. In addition, the architecture incorporates skip connections between encoder–decoder pairs of the same level to preserve high-granularity spatial information and improve the network’s overall performance, which is similar to the original UNet [10]. Thus, the network can more efficiently capture and learn complex features of the input signal, allowing for a more accurate and reliable reconstruction of the ECG signal from the PPG signal.

During training, the proposed network was fine-tuned to minimize a loss function that quantifies the dissimilarity between the network output and the actual ECG signal. The ultimate goal of the training was to optimize the weights in the network layers, to minimize this difference, and to enhance the accuracy of the PPG-to-ECG signal mapping. We employed the Adam optimizer, which adjusts the network weights based on the loss function’s gradient descent.

## 4. Experiments and Results

This study evaluated the proposed method using two training configurations:1.Group Model (GM): We generated a model using the training data from all individuals; that is, we used a generic model to capture the relationship between the PPG and ECG signals for a group of individuals.2.Subject-Specific Model (SM): A model was trained and tested for each individual to obtain a subject-specific model.

In the generalist model, the dataset was partitioned at an 80:20 ratio for training and testing. The model was trained on data from all subjects, with the exclusion of the subject selected for testing. This approach was taken to ensure that the test results were based on unseen data and that the model’s performance would be well generalized for new patients.

To generate subject-specific models, a hybrid approach was adopted. Similar to the generalist approach, the dataset was split into an 80:20 train–test ratio. However, given the limited amount of data per subject, the training set was composed of 80% of the specific subject’s data and 20% of data from other subjects. The goal of this approach was to create a more precise model tailored to the unique characteristics of the patient in question while also benefiting from the inclusion of data from other patients to enhance the overall effectiveness of the model. This approach is beneficial when the data from the patient in question are limited or the model needs to be trained on real-time data as it is collected.

The proposed models were compared with two recent methods, CardioGAN [6] and Transformers Networks [20], which are described in Section 2.

### 4.1. Training Method

During the training process, the network was initialized with random weights. The model was trained for 300 epochs, using a batch size of 32 and the Adam optimizer with a learning rate set to 0.0001. Checkpoints and early stopping were employed, with a waiting period of five epochs. The checkpoint function allowed the neural network weights to be periodically saved during training. At the same time, the early stopping method terminated training prematurely if there was no improvement in loss after a certain number of epochs. The model was developed and implemented using Python 3.8, along with the Keras 2.8.0 and TensorFlow 2.8.0 frameworks.The experiments were conducted on Google Colaboratory using GPUs of type NVIDIA Tesla T4 (or another GPU type if applicable), which facilitated faster training and processing of the neural network.

### 4.2. Evaluation Metrics

The evaluation of the similarity and dissimilarity between the signals was performed using the following metrics:

#### 4.2.1. Euclidean Distance

This measures the distance between two points in an n-dimensional space:(5)$$Euclidean\left(x,y\right)=\sqrt{{\left(x_{1}-y_{1}\right)}^{2}+\cdots +{\left(x_{n}-y_{n}\right)}^{2}}$$
where *x* and *y* are vectors of length *n*, and $x_{i}$ and $y_{i}$ are the corresponding elements of each vector.

#### 4.2.2. DTW Distance

This is a more complex metric considering the temporal deformation between two signals. It finds an optimal mapping between the points of the two signals, allowing the comparison to be performed even if the signals are out of sync:(6)$$DTW\left(x,y\right)=\sqrt{\left(min\_cost\_path{\left(x,y\right)}^{2}\right)}$$
where *x* and *y* are the time series being compared, and min_cost_path(x,y) is the path with the lowest cost between the two series.

#### 4.2.3. Pearson Correlation Coefficient

This measures the linear relationship between two variables and is commonly used to measure the similarity between two signals.
(7)$$Pearson\left(x,y\right)=\frac{cov(x,y)}{\left(std(x)*std(y)\right)}$$
where, cov(x,y) is the covariance between *x* and *y*, and std(x) and std(y) are the standard deviations of *x* and *y*, respectively.

#### 4.2.4. Spearman Correlation Coefficient

This measures the relationship between two variables based on their ranks instead of their actual values. It is a non-parametric correlation measure and is often used when the data do not have a normal distribution.
(8)$$Spearman=1-\frac{6\sum_{i=1}^{n}d_{i}^{2}}{n(n^{2}-1)}$$
where *n* is the length of the signals, and $d_{i}$ is the difference between the ranks of $x_{i}$ and $y_{i}$.

#### 4.2.5. Mean Squared Error (MSE)

This is a loss function that measures the difference between a signal generated by an algorithm and a reference signal. It is the average of the squared differences between each element of the two signals.
(9)$$MSE=\frac{1}{n}\sum_{i=1}^{n}{\left(x_{i}-y_{i}\right)}^{2}$$
where *n* is the length of the signals.

#### 4.2.6. Mean Absolute Error (MAE)

This is a measure of the average absolute difference between a model’s actual and predicted values. It measures the average size of the error in units of the data.
(10)$$MAE=\frac{1}{n}\sum_{i=1}^{n}\left|x_{i}-y_{i}\right|$$
where *n* is the length of the signals.

### 4.3. Results and Discussions

#### 4.3.1. Group Model (GM)

This evaluation aims to test the proposed method’s effectiveness in reconstructing ECG signals for multiple individuals. Figure 4 presents the reconstruction of 4 s segments of ECG signals from different users for each database.

Based on the obtained results, we can observe that the proposed method better approximated the signals in the BIDMC and CAPNO databases than the DALIA and WESAD databases. This difference in performance can be explained by the wide variation of signals collected from wearable devices in the latter databases, resulting in more significant variability in signal morphology among patients.

The increased signal variability may have made it challenging for the model to generalize to these databases, indicating the need for improvements to handle this greater signal diversity. The increased signal variability may have made it challenging for the model to generalize to these databases, indicating the need for improvements to handle this greater signal diversity.

Table 1 compares the performance of different signal mapping methods using the group model (GM) evaluation strategy. The metrics shown represent the average values calculated across all users in each dataset. The distance and similarity metrics were used to quantify the differences and similarities between the signals, validating the results of the ECG signal reconstruction for the entire user population.

Figure 4 shows the reconstruction of 4 s segments of ECG signals from randomly selected users in each of the evaluated databases (BIDMC, CAPNO, DALIA, and WESAD) using the PPG2ECG method with the generalized model. Each ECG sample corresponds to a single user per dataset. Additionally, the figure shows the PPG signal used to reconstruct the waveform of the ECG signal.

Among the evaluated methods, the PPG2ECG method showed comparable results to the others, with average metrics ranging from 3.507 to 2.169. The Euclidean metric, which measures the distance between signals, recorded higher values for the PPG2ECG method, ranging from 3.504 to 4.016. These values indicate a greater dispersion than the reference signals compared to the other evaluated methods.

However, it is interesting to note that the PPG2ECG method obtained the best averages for the DTW, Person, and Spearman metrics in all databases, indicating a more significant similarity between the signals reconstructed by this method and the original signals. Specifically, the highlight was the Pearson correlation metric with a value of 0.018, indicating a positive correlation between the mapped signals and the reference signals. These results suggest that the PPG2ECG method preserves essential aspects of the ECG signal, despite the differences being observed in the Euclidean metric. Therefore, despite a discrepancy in Euclidean distance, the PPG2ECG method demonstrated a more faithful and accurate reconstruction concerning the global and linear characteristics of the ECG signals.

#### 4.3.2. Hybrid Personalized Model

The objective of the evaluation was to test the ability of the PPG2ECG method to infer ECG signals from the corresponding PPG signals for a specific user compared to reference methods. The method was evaluated in its capability to map the PPG signal to the ECG signal, considering that the model has prior information about the correlation between these signals for a specific user. Figure 5 presents the results obtained by the PPG2ECG method when mapping a sample from each evaluated database. This evaluation was vital to assess the effectiveness and quality of the method compared to reference methods, identifying potential limitations and opportunities for improvement.

Figure 5 presents the reconstruction of 4-second segments of ECG signals from randomly selected users for each evaluated database. Each ECG sample shown corresponds to a single user per dataset. Thus, the figure illustrates a single sample per database, providing a visual example of the method’s ability to infer ECG signals from PPG for each of these users. The proposed method demonstrated more accurate results in inferring ECG signals that were similar to the original ECG when using the BIDMC and CAPNO databases. It is important to note that, for these databases, the user data used to train and test the inference models were collected in hospital environments where users were at rest or under anesthesia, resulting in more stable signals with minimal morphology variation. However, the evaluation of the hybrid personalized model for the DALIA and WESAD databases presented more challenges due to the significant variation in the signals, as they were collected during various daily activities.

Figure 6 shows the reconstruction of an ECG signal from a randomly selected patient for each evaluated comparison method. The same PPG sample from this user was used across all three methods tested. This allowed for a direct comparison of the performance of each method in reconstructing the ECG signal for the same user, ensuring consistency in the evaluation process.

The hybrid personalized approach achieved approximate reconstructions of the original signal for both methods, with fewer difficulties than the generalist approach. Table 2 compares the proposed and reference methods for ECG signal reconstruction using a hybrid personalized approach. The metrics presented in the table represent the average values calculated across all subjects, reflecting the overall performance of each method in reconstructing ECG signals under this evaluation strategy. It can be observed that PPG2ECG demonstrated similar performance to the other methods. The evaluation metrics include Euclidean distance, DTW, Pearson correlation, Spearman correlation, MSE, and MAE. The average metrics for the proposed method, PPG2ECG, were 2.680 for the Euclidean distance, 1.559 for the DTW, 0.329 for the Pearson correlation, 0.334 for the Spearman correlation, 0.015 for the MSE, and 0.073 for the MAE. These values indicate that the proposed method exhibited a lower average error in comparing the reconstructed ECG signal and the original ECG signal.

The table indicates that the proposed PPG2ECG method performed well compared to the reference methods for ECG signal reconstruction from PPG signals. The metrics demonstrate that PPG2ECG can accurately reconstruct the ECG signal, yielding results comparable to the reference methods.

## 5. Conclusions

In this article, we proposed the PPG2ECG method, which utilizes the IncResU-Net architecture to reconstruct ECG signals based on PPG signals. The results demonstrated the effectiveness of the PPG2ECG method in accurately reconstructing the original ECG signal, surpassing the performance of the compared methods.

These findings highlight the potential of deep learning architectures in signal inference. However, a more extensive and diverse ECG and PPG dataset is required for broader generalization, mainly encompassing patients with anomalous signals. This will enable better adaptation to patients’ diverse ECG morphology profiles.

Moreover, the personalized hybrid approach exhibited superior efficiency in ECG signal reconstruction, showcasing lower reconstruction errors compared to the generalized approach. Integrating machine learning techniques with this personalized approach can further enhance the reconstruction of ECG signals.

For future work, it is essential to validate the PPG2ECG method on a more extensive and diverse clinical dataset, encompassing patients with various cardiac conditions and ECG morphologies, to enhance the model’s generalization capability. Additionally, developing compression and optimization techniques for the model is crucial to optimize its suitability for resource-constrained mobile devices. Exploring transfer learning approaches also holds promise in improving the generalization capacity of the PPG2ECG model. Finally, investigating the clinical application of the method in continuous and real-time monitoring scenarios is vital for practical implementation and broader clinical adoption.

## Figures and Tables

**Figure 1 sensors-24-06046-f001:**
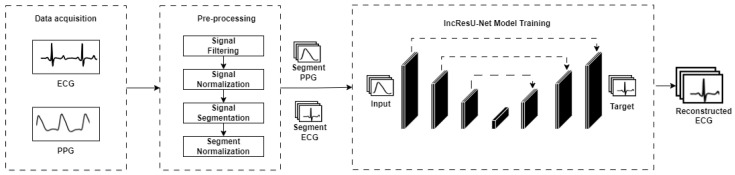
Overview of the PPG2ECG method generation process.

**Figure 2 sensors-24-06046-f002:**
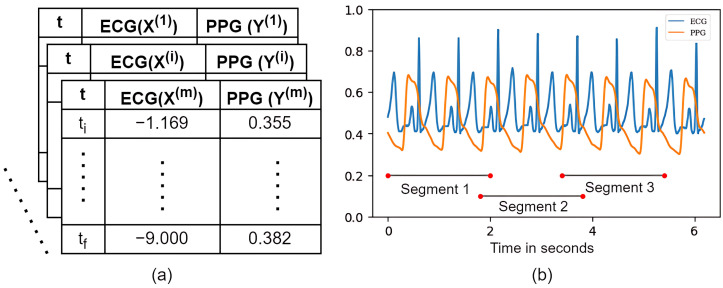
(**a**) Representation of the data after segmentation. (**b**) Graphical representation of the segmented PPG and ECG signals in overlapping windows.

**Figure 3 sensors-24-06046-f003:**
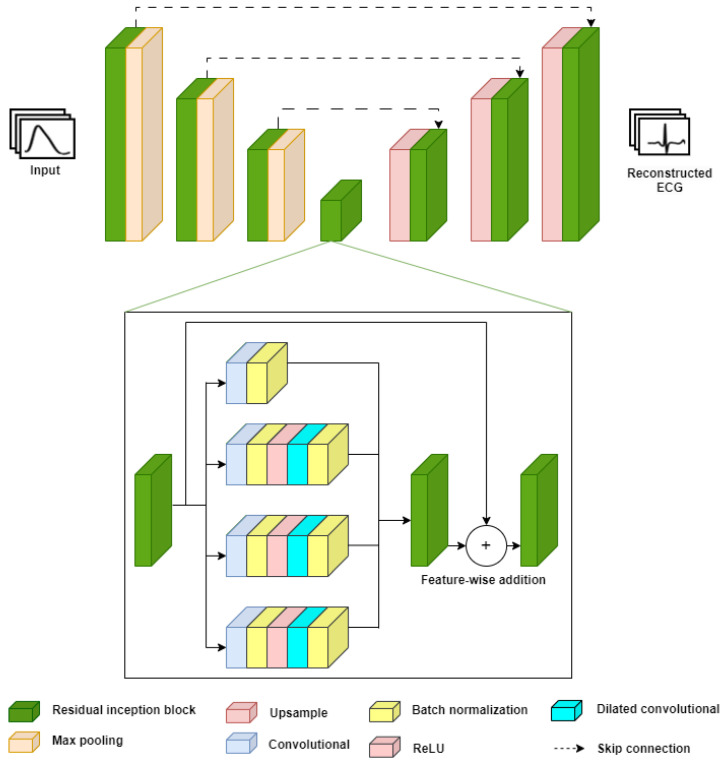
UNet Inception architecture used to generate an ECG signal.

**Figure 4 sensors-24-06046-f004:**
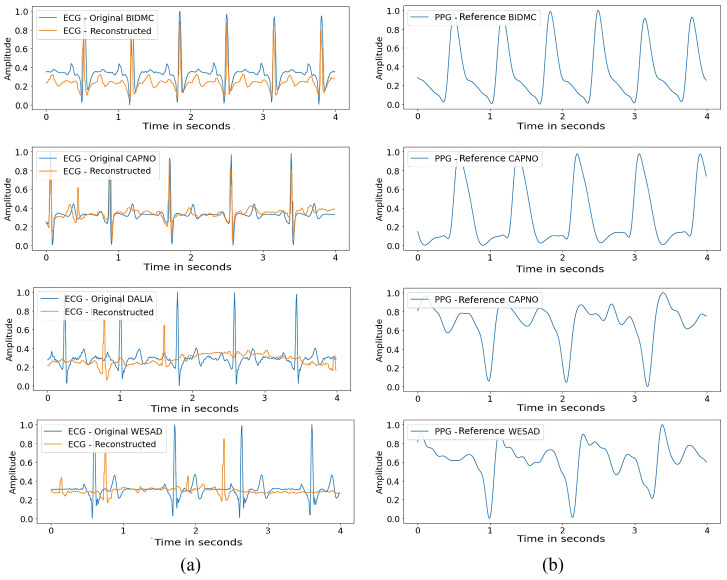
Comparison between the original ECG and the reconstructed ECG signals using the PPG2ECG method with the generalized model. (**a**) Original ECG and ECG reconstructed using the PPG2ECG method with the generalized model, and (**b**) PPG signal was used to reconstruct the waveform of the ECG signal.

**Figure 5 sensors-24-06046-f005:**
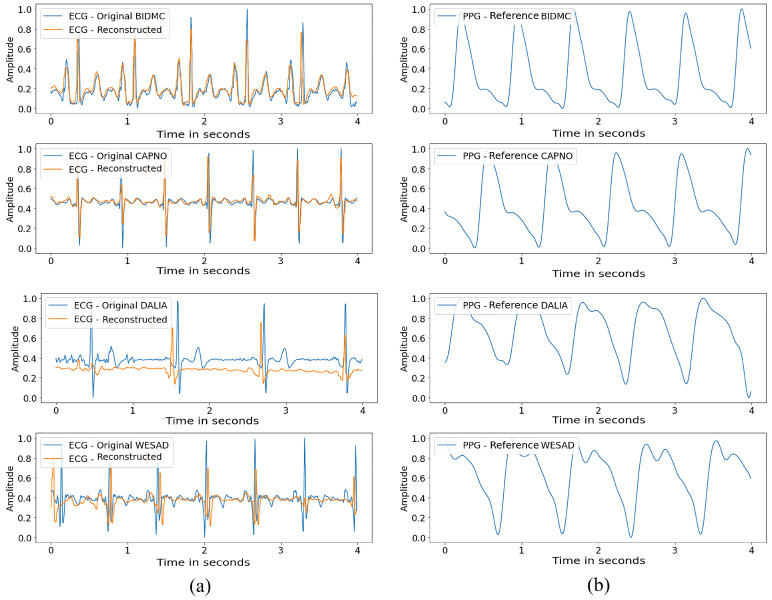
(**a**) Original ECG and ECGreconstructed using the PPG2ECG method with the personalized model, and (**b**) PPG signal was used to reconstruct the waveform of the ECG signal.

**Figure 6 sensors-24-06046-f006:**
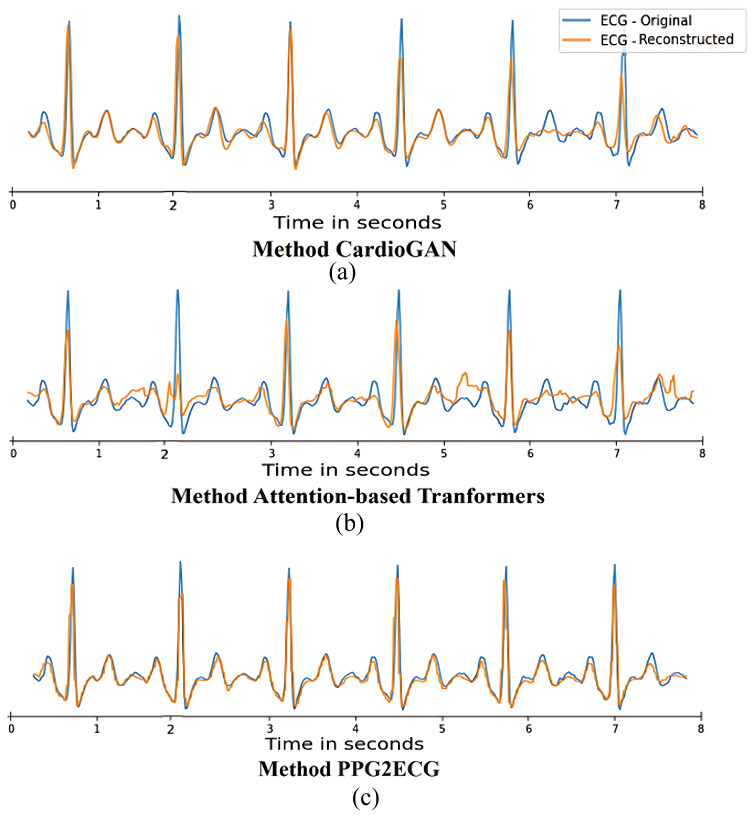
Visual comparison of the ECG reconstruction for a random user using the following methods: (**a**) CardioGAN; (**b**) Attention-based Transformers; and (**c**) Proposed PPG2ECG.

**Table 1 sensors-24-06046-t001:** Performance comparison of signal mapping methods using the group model (GM) evaluation strategy.

Method	Database	Metrics					
		**Euclidean**	**DTW**	**Pearson**	**Spearman**	**MSE**	**MAE**
CardioGan	BIDMC	4.126	2.403	0.012	0.062	0.036	0.137
	CAPNO	3.518	2.143	0.037	0.104	0.026	0.116
	DALIA	3.732	2.480	−0.001	0.000	0.028	0.132
	WESAD	2.839	1.947	−0.001	−0.002	0.016	0.081
**Average Metrics**		3.554	2.243	0.012	0.041	0.027	0.117
Attention-based Tranformers	BIDMC	4.142	2.387	0.004	−0.015	0.033	0.126
	CAPNO	3.496	2.042	0.010	0.075	0.024	0.111
	DALIA	3.424	2.835	0.001	−0.002	0.023	0.119
	WESAD	2.553	2.262	0.000	−0.003	0.012	0.062
**Average Metrics**		**3.404**	2.382	0.004	0.014	**0.023**	**0.105**
Proposed ECG2PPG	BIDMC	4.016	2.247	0.008	0.089	0.037	0.131
	CAPNO	3.504	2.022	0.064	0.102	0.025	0.114
	DALIA	3.676	2.472	0.001	0.113	0.027	0.127
	WESAD	4.016	2.247	0.008	0.089	0.037	0.131
**Average Metrics**		3.507	**2.169**	**0.018**	**0.076**	0.026	0.112

**Table 2 sensors-24-06046-t002:** Comparison of performance among signal mapping methods using the subject-specific evaluation strategy (SM).

Method	Database	Metrics					
		**Euclidean**	**DTW**	**Person**	**Spearman**	**MSE**	**MAE**
CardioGan	BIDMC	2.518	1.244	0.525	0.555	0.014	0.064
	CAPNO	1.654	1.053	0.753	0.690	0.006	0.038
	DALIA	3.321	2.003	0.004	0.007	0.022	0.104
	WESAD	3.252	2.118	0.004	0.008	0.020	0.093
**Average Metrics**		2.686	1.604	0.321	0.315	0.015	0.074
Attention-based Tranformers	BIDMC	2.924	1.751	0.366	0.360	0.018	0.082
	CAPNO	1.962	1.155	0.650	0.503	0.008	0.050
	DALIA	3.158	2.673	0.001	−0.003	0.020	0.101
	WESAD	2.865	2.529	0.001	−0.002	0.016	0.077
**Average Metrics**		2.727	2.027	0.254	0.214	0.015	0.077
Proposed ECG2PPG	BIDMC	2.482	1.177	0.559	0.601	0.013	0.057
	CAPNO	1.691	0.977	0.747	0.714	0.006	0.035
	DALIA	3.387	2.125	0.007	0.013	0.023	0.104
	WESAD	3.162	1.958	0.004	0.009	0.021	0.096
**Average Metrics**		**2.680**	**1.559**	**0.329**	**0.334**	**0.015**	**0.073**

## Data Availability

Data are contained within the article.

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
