# Peer review of "Inferring ECG Waveforms from PPG Signals with a Modified U-Net Neural Network"

_sensors, 2024, doi:10.3390/s24186046_

Round 1
Reviewer 1 Report
Comments and Suggestions for Authors
The paper proposes a method to reconstruct ECG signals from PPG signals using Convolutional Neural Network Architecture (CNN), selecting a variety of data, pre-processing them and conducting experiments. Finally a generic model and a personalized model are used to evaluate the method.
1.How exactly is the data segmented? E.g. length, segmentation points.
2.Final comparison, is it the result of single data? What are the results for multiple data?
3.The text mentions different data sources with different signal quality, and the data preprocessing is just simple filtering, is its signal usable?
Author Response
Reviewer 1
Comments and Suggestions for Authors
The paper proposes a method to reconstruct ECG signals from PPG signals using Convolutional Neural Network Architecture (CNN), selecting a variety of data, pre-processing them and conducting experiments. Finally a generic model and a personalized model are used to evaluate the method.
1.How exactly is the data segmented? E.g. length, segmentation points.
I appreciate the pertinent comments. The suggested changes have been implemented in the article. The resampling process was clarified in Section 3.1, while the duration of each segment, defined as 4 seconds, was detailed in Section 3.2.
For clarification purposes:
The data are segmented by dividing the time series into smaller segments or time windows. These segments are defined based on the signal's sampling frequency. In the study, the segmentation of the ECG and PPG signals results in a dataset X = {( x(1) , y(1)), ( x(2) , y(2)), …, ( x(m) , y(m))}, where x (i) represents the input PPG signal and y(i) represents the reference ECG signal. Each segment x(i) must be the same size as the reference signal y(i) to ensure synchronization between the signals.
Segmentation Details
- Sampling Frequency: To unify the sampling frequency among the different devices used to collect the signals, a resampling process was performed through interpolation, standardizing the signals to a frequency of 128 Hz.
- Segment Duration: Each segment has a duration of 4 seconds.
- Calculation of Points per Segment:
Number of points per segment = Sampling frequency*Segment duration
Substituting the values:
Number of points per segment=128 Hz*4 seconds=512
Overlap: To avoid losing important peaks in the signals, each segment has a 10% overlap with the previous segment.
Thus, each data segment contains 512 points and covers a 4-second interval, with a 10% overlap to ensure that critical features of the ECG and PPG signals, such as peaks, are not lost during the segmentation process. This allows for precise and synchronized analysis of the signals, which is essential for accurately reconstructing ECG signals from PPG signals.
2.Final comparison, is it the result of single data? What are the results for multiple data?
We appreciate the observation. The final comparison presented in the section uses a single user sample (e.g., Figure 6) to illustrate the ECG reconstruction results across different methods. However, the results shown in Table 2 reflect the average of multiple samples from various users for each of the evaluated datasets (BIDMC, CAPNO, DALIA, and WESAD). The performance metrics presented in the table (Euclidean distance, DTW, Pearson correlation, Spearman correlation, MSE, and MAE) are calculated from multiple data points, allowing for a more robust and representative evaluation of the PPG2ECG method's effectiveness.
We have revised the descriptions of all tables and figures to make this information clearer.
3.The text mentions different data sources with different signal quality, and the data preprocessing is just simple filtering, is its signal usable?
We acknowledge the concern regarding the data preprocessing and the quality of the signals. In our study, we utilized the Butterworth filter for preprocessing the signals, as it has been demonstrated to be the most suitable for this type of biomedical signal, particularly for ECG and PPG, as reported by Liang [26]. The Butterworth filter is well-known for its maximally flat frequency response in the passband, which minimizes signal distortion while effectively removing noise. This characteristic makes it ideal for preserving the morphology of the signals, which is crucial for accurate ECG reconstruction.

Reviewer 2 Report
Comments and Suggestions for Authors
The manuscript entitled “Inferring ECG Waveforms from PPG Signals with a Modified U-net Neural Network” is well structured and represents a thorough investigation of the reconstruction of ECG signals from PPG signals using the IncResU-Net architecture. Although the study is solid and makes an important contribution to the field, some minor revisions are needed to improve clarity and ensure accessibility to a broader audience. These revisions will improve the overall quality and readability of the manuscript.
The manuscript contains a well-structured introduction and a thorough review of related work that effectively places the study in the context of existing research on the reconstruction of ECG signals from PPG signals. However, some minor issues should be addressed to improve clarity. The citation format should be revised to combine multiple references into a single set of square brackets (e.g., [4, 5] instead of [4][5]). The abbreviation "kNN" introduced in line 116 should be explained as "k-Nearest Neighbors" to provide clarity for readers unfamiliar with the term. In addition, medical abbreviations such as RR, PR, QRS and QT intervals should be defined at their first mention to make the manuscript easier to understand for readers without a cardiology background.
The methodology section is well structured and provides a clear and comprehensive explanation of the process from data acquisition to implementation of the customized IncResU-Net architecture. However, there are some minor areas that could be improved. The abbravation "IncResU-Net" should be explained as soon as it is introduced to provide clarity for readers unfamiliar with the term, rather than later in the text. Similarly, the abbreviation "ReLU" should be explained at the first mention (rectified linear unit). Including an equation to illustrate the ReLU function would also improve the reader's understanding. According to Figure 3, the manuscript clearly describes the filter sizes (kernel sizes) and effectively conveys their role within the network. In this context, you could explain in more detail how these choices affect the performance of the model and explain the reasons for choosing certain filter sizes to make the manuscript more understandable and clear. It is also important to describe in detail the hardware used for these calculations to ensure the repeatability of your study.
The Experiments and Results section contains a comprehensive and methodological evaluation of the proposed PPG2ECG method. The dual approach to model training, which includes both a generalized and a hybrid personalized strategy, is well thought out and clearly presented. The methodology for splitting the dataset, the detailed description of the training process and the use of advanced machine learning techniques are well presented. The results are clearly presented and discussed with insightful observations. Overall, this part of the manuscript successfully demonstrates the effectiveness of the proposed models and provides a transparent evaluation of their performance, representing a significant contribution to the field of biomedical signal processing.
The conclusion of the manuscript summarizes the significant achievements of the PPG2ECG method and highlights its superior performance in ECG signal reconstruction from PPG signals using the IncResU-Net architecture. The manuscript outlines a commendable future research agenda that includes expanding the dataset to improve model generalization, improving the personalized hybrid approach, and optimizing the model for mobile use. The proactive approach to integrating advanced machine learning techniques and exploring clinical applications for real-time monitoring shows a clear path to practical implementation and broad clinical application. This section succinctly summarizes the essence of the study contributions and provides a strategic direction for further progress in this area.
The references are appropriate and relevant, including recent studies and basic research that support the methodology and results. The tables are well organized and present the data clearly, while the figures effectively illustrate key aspects of the study.
Author Response
Reviewer 2
Comments and Suggestions for Authors
Some minor issues should be addressed to improve clarity. The citation format should be revised to combine multiple references into a single set of square brackets (e.g., [4, 5] instead of [4][5]). The abbreviation "kNN" introduced in line 116 should be explained as "k-Nearest Neighbors" to provide clarity for readers unfamiliar with the term. In addition, medical abbreviations such as RR, PR, QRS and QT intervals should be defined at their first mention to make the manuscript easier to understand for readers without a cardiology background ….
Answer:
We appreciate the reviewer’s comments and suggestions. All requested changes have been made to improve the clarity, readability, and overall quality of the manuscript. Below is a list of the revisions made:
- Citation format: Multiple references have been grouped into a single set of square brackets (e.g., [4, 5] instead of [4][5]).
- Clarification of "kNN": The abbreviation "kNN" has been explained as "k-Nearest Neighbors" at its first mention for better understanding.
- Definition of medical abbreviations: Abbreviations such as RR, PR, QRS, and QT intervals have been defined at their first mention to assist readers without a cardiology background.
- Explanation of "IncResU-Net": The abbreviation "IncResU-Net" has been explained upon first mention to provide immediate clarity for readers unfamiliar with the term.
- Explanation of "ReLU": The abbreviation "ReLU" has been explained as "rectified linear unit" at its first mention, and an equation illustrating the ReLU function has been added for clarity.
- Filter size explanation: We have added a detailed explanation of how the choice of filter sizes affects the performance of the model and the reasoning behind these choices.
- Hardware description: The hardware used for model training and experiments, including the use of Google Colaboratory with NVIDIA Tesla T4 GPUs, has been described to ensure reproducibility.

Reviewer 3 Report
Comments and Suggestions for Authors
The issues addressed in the article are not clearly defined, and the challenging aspects of the problem are not well articulated.
The innovation points of this article are too general, as the proposed method is merely a simple modification of IncResU-Net.
There are too few comparative methods, which fails to demonstrate the superiority of the proposed approach.
It should clearly state what previous researchers have done, what their limitations were, and how the current study contributes new knowledge to the existing literature.
Comments on the Quality of English LanguageMinor editing of English language required.
Author Response
Reviewer 3
Comments and Suggestions for Authors
The issues addressed in the article are not clearly defined, and the challenging aspects of the problem are not well articulated.
The innovation points of this article are too general, as the proposed method is merely a simple modification of IncResU-Net.
There are too few comparative methods, which fails to demonstrate the superiority of the proposed approach.
It should clearly state what previous researchers have done, what their limitations were, and how the current study contributes new knowledge to the existing literature.
Response to Reviewer 3:
We appreciate your comments and suggestions on our manuscript. We highly value your feedback, which has helped us identify areas for improvement to enhance the clarity and quality of our work.
We agree that the issues addressed in the article could be more clearly defined. In the revised manuscript, we explicitly articulate the specific challenges in reconstructing ECG signals from PPG signals, highlighting the complexities inherent to this process, such as signal variability, the need for precise synchronization, and the preservation of ECG signal morphology during reconstruction.
We acknowledge that the innovation points need to be better explained. Although our approach utilizes a modification of the IncResU-Net, we believe that the adaptations made for this specific context represent a significant contribution. In the revised manuscript, we detail how the specific modifications to the architecture improve the model's ability to capture and accurately reconstruct ECG signals, particularly in scenarios with signal variability.
Regarding the comparative methods, we understand the importance of demonstrating the effectiveness of the proposed approach. The chosen comparison methods were selected based on their relevance and recognition in the current literature. Comparing our method with these specific approaches allowed for a rigorous and focused evaluation of our performance against established standards. We believe this comparison provides a clear view of the advantages and limitations of our method, highlighting its contributions to the field.
In the revised manuscript, we have provided a more detailed review of previous researchers' work, highlighting the limitations of those studies and clearly explaining how our study contributes new knowledge to the existing literature, advancing the state of the art in ECG reconstruction from PPG signals.
